# Post-myocardial infarction heart failure and long-term high-fat diet: Cardiac endoplasmic reticulum stress and unfolded protein response in Sprague Dawley rat model

Karol Momot[1], Kamil Krauz[1], Katarzyna Czarzasta[1], Jakub Tomaszewski[1], Jakub Dobruch[2], Tymoteusz Żera[1], Maciej Zarębiński[3], Agnieszka Cudnoch-Jędrzejewska[1], Małgorzata Wojciechowska[1]*

1 Laboratory of Centre for Preclinical Research, Department of Experimental and Clinical Physiology, Medical University of Warsaw, Warsaw, Poland, 2 Centre of Postgraduate Medical Education, Department of Urology, Warsaw, Poland, 3 Department of Invasive Cardiology, Independent Public Specialist Western Hospital John Paul II, Lazarski University, Grodzisk Mazowiecki, Poland

* malgorzata.wojciechowska2@wum.edu.pl

## Abstract

### Background

Myocardial infarction (MI) significantly contributes to the global mortality rate, often leading to heart failure (HF) due to left ventricular remodeling. Key factors in the pathomechanism of HF include nitrosative/oxidative stress, inflammation, and endoplasmic reticulum (ER) stress. Furthermore, while a high-fat diet (HFD) is known to exacerbate post-MI cardiac remodeling, its impact on these critical factors in the context of HF is not as well understood.

### Aims

This study aimed to assess the impact of post-MI HF and HFD on inflammation, nitro-oxidative stress, ER stress, and unfolded protein response (UPR).

### Methods

The study was performed on fragments of the left ventricle harvested from 30 male adult Sprague Dawley rats, which were divided into four groups based on diet (normal-fat vs. high-fat) and surgical procedure (sham operation vs. coronary artery ligation to induce MI). We assessed body weight, NT-proBNP levels, protein levels related to nitrosative/oxidative stress, ER stress, UPR, apoptosis, and nitric oxide synthases, through Western Blot and ELISA.

### Results

HFD and MI significantly influenced body weight and NT-proBNP concentrations. HFD elevated 3-nitrotyrosine and myeloperoxidase levels and altered nitric oxide synthase levels. HFD and MI significantly affected ER stress markers and activated or inhibited UPR pathways.

**Data Availability Statement:** All relevant data are within the manuscript.

**Funding:** This study was financed by a research grant from the Medical University of Warsaw (1MA/2/M/MG/N/23) to [KK] https://pnitt.wum.edu.pl/en The funders had no role in study design, data collection and analysis, decision to publish, or preparation of the manuscript.

**Competing interests:** The authors have declared that no competing interests exist.

## Conclusions

The study demonstrates significant impacts of post-MI HF and dietary fat content on cardiac function and stress markers in a rat model. The interaction between HFD and MI on UPR activation suggests the importance of dietary management in post-MI recovery and HF prevention.

## Introduction

Myocardial infarction (MI) is a significant cause of death worldwide [1]. Due to modern treatment options, the mortality from MI has been decreasing, resulting in a growing population of MI survivors [2]. Many of these individuals subsequently develop symptoms of heart failure (HF) [3, 4]. After the cardiomyocytes' death due to ischemia, the development of HF is related to unfavorable left ventricular remodeling, leading to loss of function [5, 6]. Consumption of a high-fat diet (HFD) can intensify the remodeling after MI through mechanisms such as cardiac hypertrophy, cardiomyocyte apoptosis and interstitial fibrosis [7, 8].

Experimental studies have shown that HFD significantly exacerbates hypertensive heart disease in aging rats, leading to worsened atrial and ventricular remodeling and associated impairment of left ventricular systolic function [9]. Moreover, just 12 weeks of HFD can adversely affect cardiac function, as measured by the left ventricular speckle tracking imaging [10], a parameter capable of detecting subclinical left ventricular. Unfortunately, recent clinical studies have revealed that the consumption of high-fat products in the human population has been steadily increasing [11]. In the context of HF, there's much discussion about nitrosative/oxidative stress, inflammation, and ER stress [12–15]. However, little is known about the impact of HFD on these processes in HF.

Nitrosative/oxidative stress refers to the biochemical reaction between nitric oxide (NO) and reactive oxygen species when a disorder in oxygen metabolism is present. This process results in the generation of reactive nitrogen species (such as the peroxynitrite anion), which cause nitration and damage to proteins [16]. A marker of such a damage is the 3-nitrotyrosine (3-NT) [17].

Production of NO is catalyzed by NO synthase (NOS), which has three isoforms: inducible NOS (iNOS), endothelial NOS (eNOS), and neuronal NOS (nNOS) [18]. These isoforms play crucial roles in cardiovascular health and disease. iNOS is expressed in normal heart tissue at very low levels [19]. Inflammation results in iNOS activation and overexpression, which is linked with harmful effects on the heart, whereas overexpression of nNOS and eNOS in transgenic animals improves cardiac functions following MI [20]. Myeloperoxidase (MPO) plays a vital role in the inflammatory response [21]. It is expressed mainly in neutrophils and monocytes. MPO catalyzes the production of hypochlorous acid, a potent oxidative agent [22]. Moreover, this protein can also directly contribute to forming reactive nitrogen species. Enhanced circulating levels of MPO are related to inflammation and oxidative stress [23]. What is more, a recent meta-analysis suggests that MPO can be a valuable marker for HF diagnosis [24].

ER stress occurs when misfolded or unfolded proteins overwhelm the ER, a crucial cell organelle for protein folding and lipid biosynthesis. As previously mentioned, nitrosative/oxidative stress affects the protein folding process and contributes to ER stress [25, 26]. The latter activates the unfolded protein response (UPR), a complex signaling network aiming to restore proteostasis or promote apoptosis when it is not possible. This process is crucial in the

pathogenesis of HF [13]. Especially, UPR, ER stress, and iNOS overexpression have been found to influence heart failure with preserved ejection fraction (HFpEF) pathogenesis [12].

In physiological conditions, when ER stress is not exacerbated, 78-kDa glucose-regulated protein (GRP78) is attached to the ER stress sensors—inositol-requiring enzyme type 1 α (IRE1α), activating transcription factor 6 (ATF6), protein kinase R-like endoplasmic reticulum kinase (PERK) and, thus they remain inactive. When unfolded proteins excessively accumulate in the ER and ER stress exacerbates, GRP78 dissociates from these sensors, which activates them and initiates downstream signaling pathways. When ATF6 is released from GRP78, it migrates to the Golgi apparatus, where it is cleaved. Then, the cleaved ATF6 (ATF6c) is translocated to the nucleus and functions as an active transcription factor [27]. If severe ER stress persists, apoptotic pathways are activated. It has been reported that overexpression of CHOP leads to apoptosis due to ER stress [28].

This study aimed to assess the impact of post-MI HF and HFD on inflammation, nitro-oxidative stress, ER stress, and UPR.

## Methods

### Experimental procedures

The study was performed on fragments of the left ventricle harvested from 30 male adult Sprague Dawley rats, which underwent the following procedures (Fig 1, Table 1). This study is a continuation of previous experiments [29–31]. All experiments were approved by the Second Local Animal Research Ethics Committee of the Medical University of Warsaw and followed the European Communities Council Directive 2010/63/E.U. Rules of September 2010.

Starting from the fourth week of age, the animals were fed with HFD (31% fat, 17.1% protein, 35.5% carbohydrates, 0.18% sodium, and 3842 kcal/kg; Labofeed B, Kcynia, Poland) or a normal fat diet (NFD) (3.6% fat, 17.4% protein, 60% carbohydrates, 0.2% sodium, and 2864 kcal/kg; Labofeed B, Kcynia, Poland) for twelve weeks until the sixteenth week of age. The detailed composition of NFD and HFD is provided in Table 2. At the twelfth week of age, the surgical procedures were performed. During the surgeries, animals were under general anesthesia (Ketamine 10mg/100g body weight i.p., Xylazine 1mg/100g body weight i.p.). The MI model consists of a permanent ligation of the left coronary artery (LCA) with a suture thread (Ethicon 6.0). The sham surgeries (SO) were similar, but the pericardium was only touched with a needle, and the LCA was not ligated. Following surgery, the animals were administered analgesic medication (Buprenorphine chloride 3μg/100g body weight i.p.; 5.95nmol/ml, twice daily for 2–3 days) and an antibiotic (Penicillin, 10,000 IU/100g body weight i.m.; 0.047mmol/ml).

Four weeks following surgery were given to develop the post-MI HF in rats with the LCA ligation. After this period, the rats were anesthetized again. A 4 ml blood sample was taken

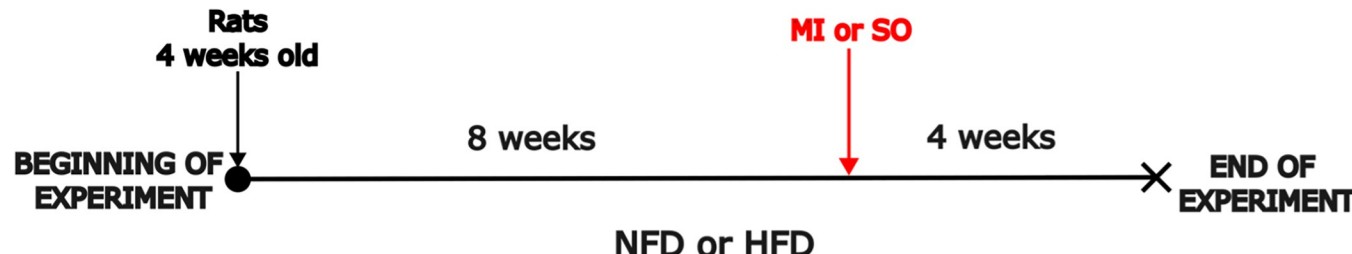

**Fig 1. Graphical representation of the study design.** Abbreviations: HFD, high-fat diet; MI, myocardial infarction; NFD, normal-fat diet; SO, sham surgery.

**Table 1. The experimental rat groups.**

| | Group 1 N = 7 | Group 2 N = 9 | Group 3 N = 6 | Group 4 N = 8 |
|---|---|---|---|---|
| Diet | Normal-fat Diet (NFD) | Normal-fat Diet (NFD) | High-fat Diet (HFD) | High-fat Diet (HFD) |
| Procedure | Sham operation (SO) | Ligation of the coronary artery (MI) | Sham operation (SO) | Ligation of the coronary artery (MI) |
| Survival rates at the end of the study | 70% (7 of 10) | 60% (9 of 15) | 60% (6 of 10) | 50% (8 of 16) |

from the right ventricle. The collected blood was immediately transferred into tubes containing Ethylenediaminetetraacetic acid (EDTA). The tubes were then centrifuged to separate the plasma from other blood components, and then levels of N-terminal prohormone of brain natriuretic peptide (NT-proBNP) were assessed using an enzyme-linked immunosorbent assay (ELISA).

At the end of the experiments, the rats were euthanized with an intraperitoneal injection of a lethal dose of Ketamine (300mg/100g body weight i.p.). The heart muscle tissue of the left ventricle was isolated, frozen in liquid nitrogen, and then stored at −80°C.

## Protein analysis

Collected tissues were homogenized using the RIPA buffer, whose composition was previously described [29]. Samples containing 10μg/μl of total protein were resolved by 8% SDS-polyacrylamide gels. Isolated proteins were transferred into PVDF membranes (#1704274, Bio-Rad) or nitrocellulose membranes using the Trans-Blot® TurboTM Transfer System (Bio-Rad). Then, blots were blocked with 5% nonfat dry milk buffered solution, immunoblotted for one hour with anti-IRE1α (NB100-2324, Novus-Biologicals), anti-p-IRE1α (NB100-2323), anti-iNOS (NB300-605), anti-eNOS (NB300-500), anti-nNOS (NBP1-39681), anti-GRP78 (NBP1-06274), anti-ATF6 (NBP1-76675), anti-PERK (NBP3-12891), anti-CHOP (NBP2-13172), anti-3NT (sc-32757, Santa Cruz Biotechnology), and anti-MPO (MPO-101AP, Thermo Fisher Scientific) antibodies. As a loading control, an anti-beta actin antibody was used (ab8226, Abcam). Incubation with secondary antibodies conjugated to horseradish peroxidase (ab205718, Abcam or sc-516102, Santa Cruz) was performed for one hour. The specific bands were detected and quantified with the ChemiDoc MP Imaginating System (Bio-Rad), and then protein expressions were normalized with β-actin and expressed as a relative ratio. Each measurement was repeated three times, and the final result was calculated as the average of these repetitions. The raw data images are provided as Supporting Information (S1 Raw images).

**Table 2. Composition of diets.**

| | Normal-fat Diet | High-fat Diet |
|---|---|---|
| Energy (kcal per 100g) | 286 | 382 |
| Carbohydrates (g per 100g) | 60.0 | 35.5 |
| Proteins (g per 100g) | 17.4 | 17.1 |
| Fats (g per 100g) | 3.2 | 28.0 |
| Saturated (%) | 13 | 49 |
| Unsaturated (%) | 37 | 44 |
| Polyunsaturated (%) | 50 | 7 |
| Raw ash (g per 100g) | 12 | 12 |
| Water (g per 100g) | 12 | 12 |

## Statistical analysis

Statistical analysis was carried out with the Statistica software, version 13.3. Normal distribution was tested using the Shapiro-Wilk test. The Levene test was used to assess the equality of variances. A two-way ANOVA was conducted to examine the effects of diet (Normal-fat Diet vs. High-fat Diet) and procedure (Sham operation vs. Myocardial infarction), as well as their interaction. Post-hoc comparisons were performed using Tukey's HSD test to identify significant differences between groups. For all the variables measured, outliers that were 1.5 interquartile ranges (IQRs) below the first quartile or 1.5 IQRs above the third quartile were removed from the analysis. Pearson's correlation coefficient was calculated to assess linear relationships between variables. Box plots were created using BioRender.com, and scatter plots with regression lines illustrating the relationships between variables were generated using an AI-based tool. All values presented in the text and figures are expressed as mean ± standard deviation (SD). All differences were considered significant if P < 0.05.

## Results

### Basic parameters

At the end of the experiment, significant differences in body weight were observed among the groups (Fig 2A). Notably, the group with post-MI HF, on an NFD, had the lowest body mass compared to other groups. A simple analysis of the main effects indicated that post-MI HF and diet significantly influenced the final body weight (P = 0.028 and P = 0.031, respectively).

NT-proBNP circulating levels differed significantly between groups (Fig 2B). Notably, higher levels of NT-proBNP were observed in groups with post-MI HF–NFD-MI and HFD-MI. A simple main effects analysis revealed that post-MI HF alone had a significant effect on serum NT-proBNP levels (P < 0.0001).

### Nitrosative/oxidative stress and inflammation

The levels of 3-NT in the left ventricle tissue showed significant differences between the rat groups (Fig 3A). The group with post-MI HF and an implemented HFD exhibited the highest

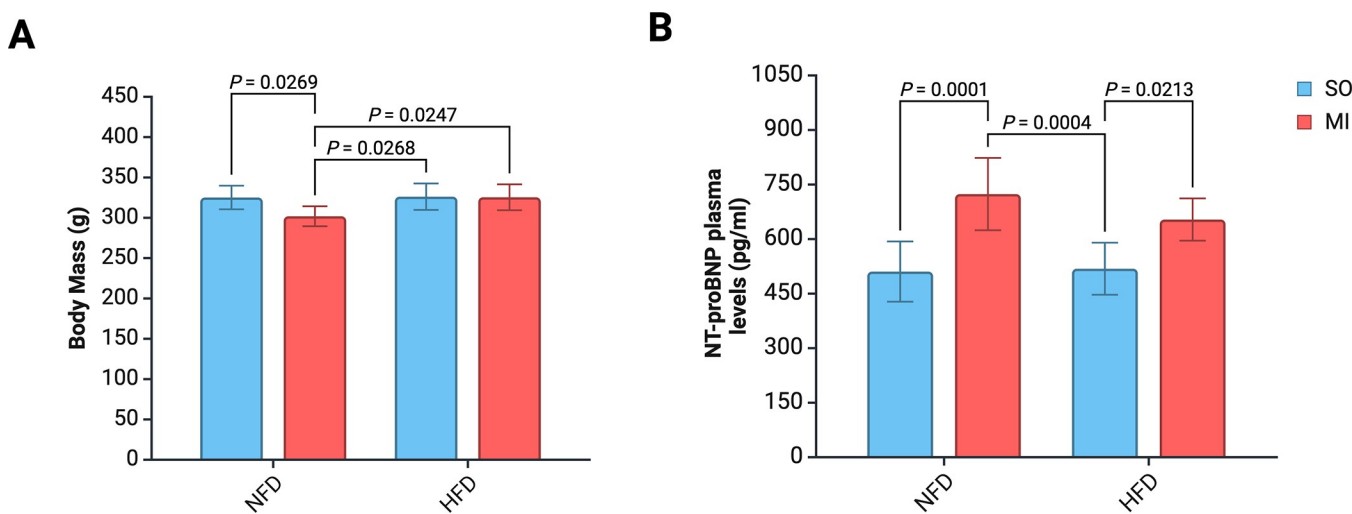

**Fig 2. Basic parameters at the end of experiments.** Abbreviations: HF, heart failure; HFD, high-fat diet; NFD, normal-fat diet; SO, sham surgery.

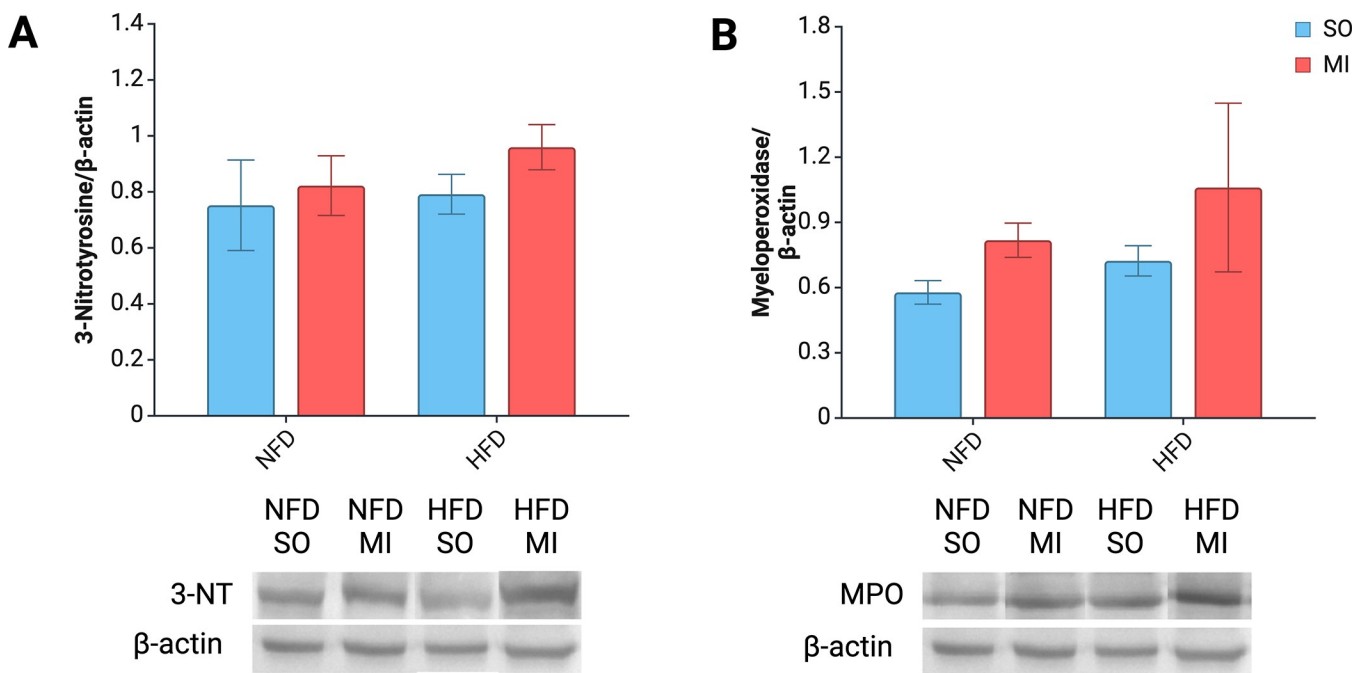

**Fig 3. Bar graphs and representative Western blot images.** 3-nitrotyrosine (A) and myeloperoxidase (B). Abbreviations: 3-NT, 3-nitrotyrosine; HF, heart failure; HFD, high-fat diet; MPO, myeloperoxidase; NFD, normal-fat diet; SO, sham surgery.

3-NT levels. A simple main effects analysis showed that post-MI HF and HFD had a significant effect on 3-nitrotyrosine levels (P = 0.021 and P = 0.048, respectively).

The levels of MPO in the left ventricular tissue showed significant differences among the groups (Fig 3B). The group that developed HF after MI and was subjected to HFD exhibited the highest levels of MPO. A two-way ANOVA indicated that post-MI HF and diet significantly impacted MPO levels in the left ventricular tissue (P = 0.004 and P = 0.028, respectively).

## Nitric oxide synthases levels

Levels of nNOS in the left ventricular tissue significantly differed between groups (Fig 4A). Simple main effects analysis showed that post-MI HF did have a significant effect on nNOS levels (P < 0.001). It was associated with reduced levels of eNOS independently of the type of diet implemented.

There was a significant difference in eNOS levels within the left ventricular tissue between groups (Fig 4B). eNOS levels were significantly reduced in other groups compared to the control group–NFD-SO. A two-way ANOVA revealed a significant interaction between the effects of post-MI HF and diet (F(1,25) = 5.995, P = 0.022).

The levels of iNOS in the left ventricle tissue showed significant differences between the groups (Fig 4C). The rats with post-MI HF and an implemented HFD exhibited the highest iNOS levels, significantly higher than other groups. Simple main effects analysis showed that both post-MI HF and diet did have a significant effect on iNOS levels (P < 0.001 and P = 0.010, respectively).

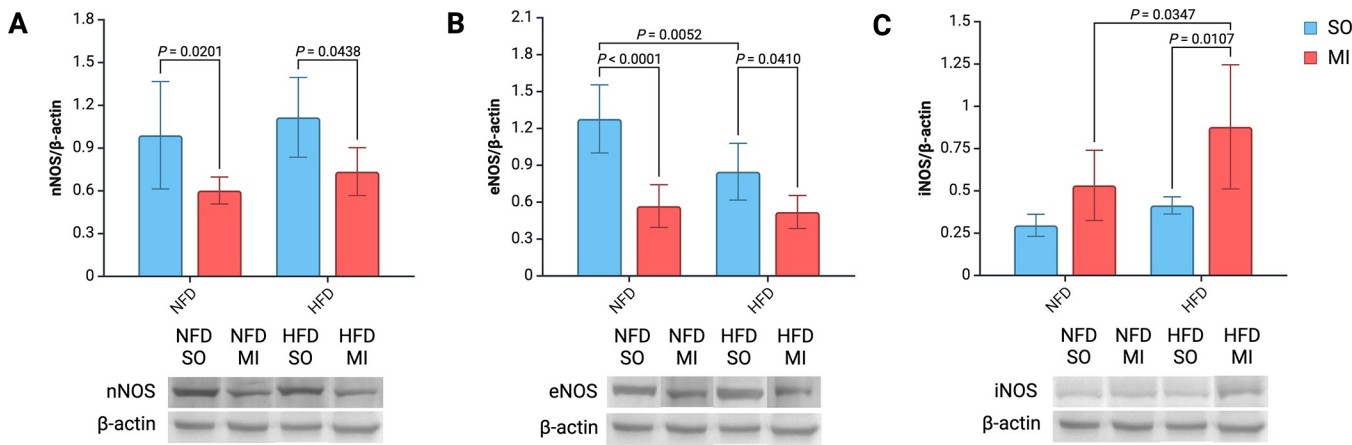

**Fig 4. Bar graphs and representative Western blot images.** nNOS (A), eNOS (B) and iNOS (C). Abbreviations: 3-NT, 3-nitrotyrosine; HF, heart failure; HFD, high-fat diet; MPO, myeloperoxidase; NFD, normal-fat diet; SO, sham surgery; eNOS, endothelial nitric oxide synthase; iNOS, inducible nitric oxide synthase; nNOS, neuronal nitric oxide synthase.

### Endoplasmic reticulum stress, unfolded protein response, and apoptosis

There was a significant difference in GRP78 protein levels within left ventricular tissue among the various groups (Fig 5A). GRP78 levels were significantly higher in rats subjected to HFD than those on NFD across all combinations. Furthermore, a simple main effects analysis revealed that only diet statistically impacted GRP78 levels ($P < 0.0001$).

The activity of the IRE1α axis of the UPR, measured by the ratio of phosphorylated IRE1α to non-phosphorylated IRE1α, showed significant variation between groups (Fig 5B). The lowest activity was observed in rats subjected to HFD and after MI. A simple main effects analysis revealed that only diet affected the p-IRE1α/IRE1α ratio ($P = 0.026$). A two-way ANOVA indicated that the interaction between the effects of post-MI HF and diet on the activity of the IRE1α axis was significant ($F_{(1, 22)} = 7.401$, $P = 0.012$).

The activity of the ATF6 axis of the UPR, measured by the ratio of cleaved ATF6 to non-cleaved ATF6, showed significant differences between groups (Fig 5C). The highest activity was observed in rats subjected to HFD after MI. This activity level was significantly higher than in other groups. Both post-MI HF and diet affected the activity of the ATF6 axis ($P < 0.0001$ and $P < 0.0001$). Furthermore, a two-way ANOVA revealed a significant interaction between the effects of post-MI HF and diet on the activity of the ATF6 axis ($F_{(1, 24)} = 8.707$, $P = 0.007$).

PERK levels in the left ventricular tissue significantly differed between groups. The rats subjected to an NFD without post-MI HF exhibited the highest levels of PERK (Fig 5D). This difference was significant compared to other groups. Other groups showed similar levels of PERK, with no significant differences. A two-way ANOVA indicated a significant interaction between the effects of post-MI HF and diet on PERK levels ($F_{(1, 25)} = 7.988$, $P = 0.009$).

CHOP levels in left ventricular tissue varied significantly between groups (Fig 5E). The highest level of CHOP was observed in the control group (NFD-SO). This difference was significant compared to other groups. Additionally, a two-way ANOVA demonstrated that post-MI HF and diet significantly affected CHOP levels in heart left ventricular tissue ($P < 0.0001$ and $P < 0.001$, respectively).

As depicted in Fig 6, the levels of 3-nitrotyrosine correlated with GRP78 levels ($\rho = 0.571$, $P = 0.002$).

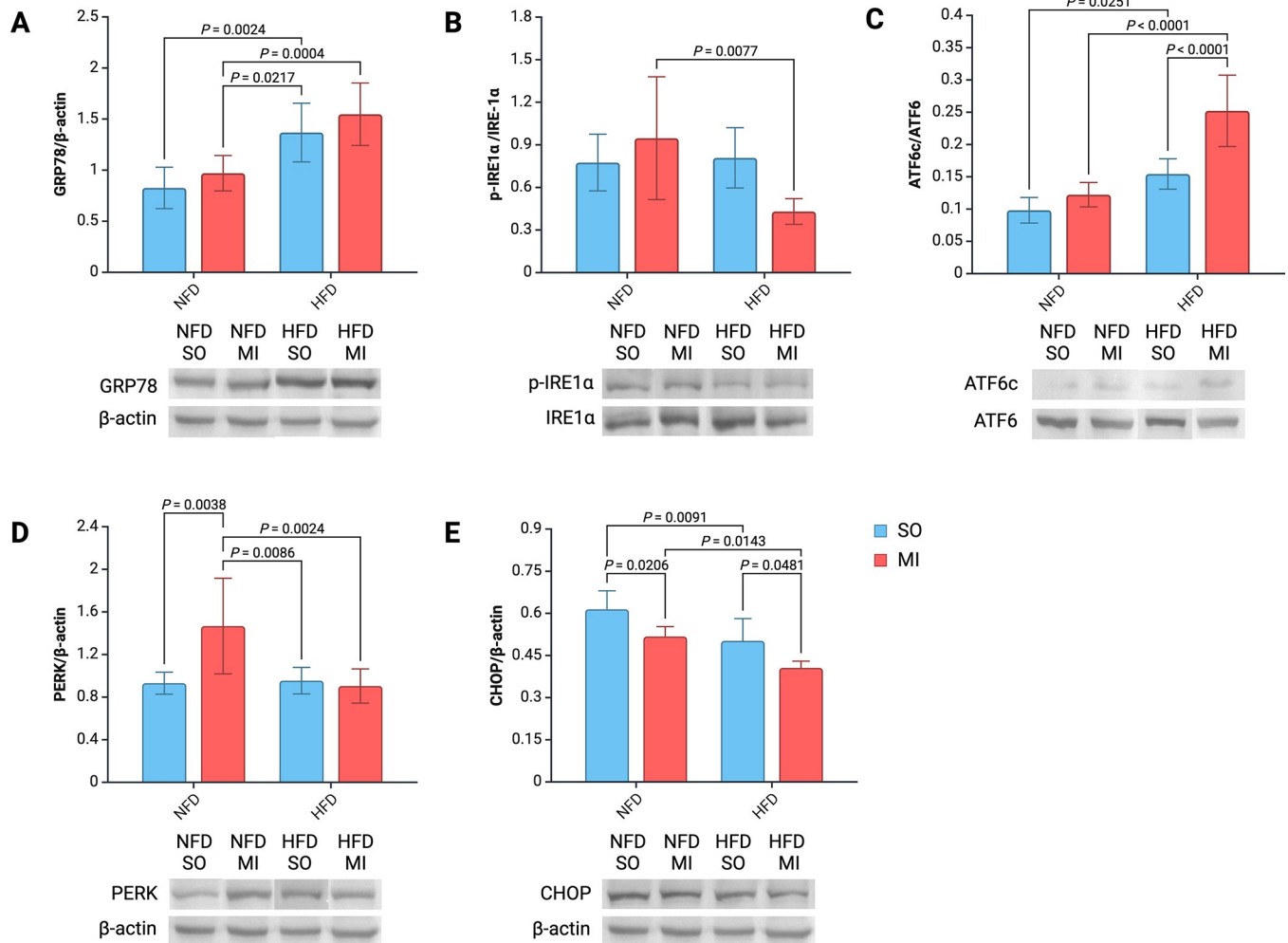

**Fig 5. Bar graphs and representative Western blot images.** GRP78 (A), p-IRE1α/IRE1α (B), ATF6c/ATF6 (C), PERK (D) and CHOP (E). Abbreviations: ATF6, activating transcription factor 6; ATF6c, activating transcription factor 6 (cleaved form); CHOP, C/EBP homologous protein; GRP78, 78-kDa glucose-regulated protein; HF, heart failure; HFD, high-fat diet; IRE1α, inositol-requiring enzyme type 1 α; NFD, normal-fat diet; PERK, protein kinase R-like endoplasmic reticulum kinase; SO, sham surgery; p-IRE1α, phosphorylated inositol-requiring enzyme type 1 α.

## Discussion

This preliminary study sheds light on the pathogenesis of post-MI HF and its exacerbation by HFD. We have demonstrated that in our model of post-MI HF with an implemented HFD, we can observe disturbances similar to those seen in HFpEF—specifically ER stress, nitrosative/oxidative stress, and disturbances in the UPR. Given that HFpEF is typically not associated with MI, this finding offers a new perspective on the mechanisms underlying HF development and progression.

Our study found that HFD was associated with increased ER stress measured by GRP78 levels. Zhang et al. also revealed that GRP78 was highly expressed in the atrial myocardium of HFD mice [32]. Additionally, they demonstrated that GRP78 expression in the atrial myocardium of overweight patients was significantly higher. These data suggest that ER stress could be activated in the myocardium not only in the case of obesity but also in HFD. Interestingly, in our study, post-MI HF had no significant impact on GRP78 levels. Contrary to our findings, Mainali et al. demonstrated that after inducing MI, GRP78 was markedly upregulated in heart

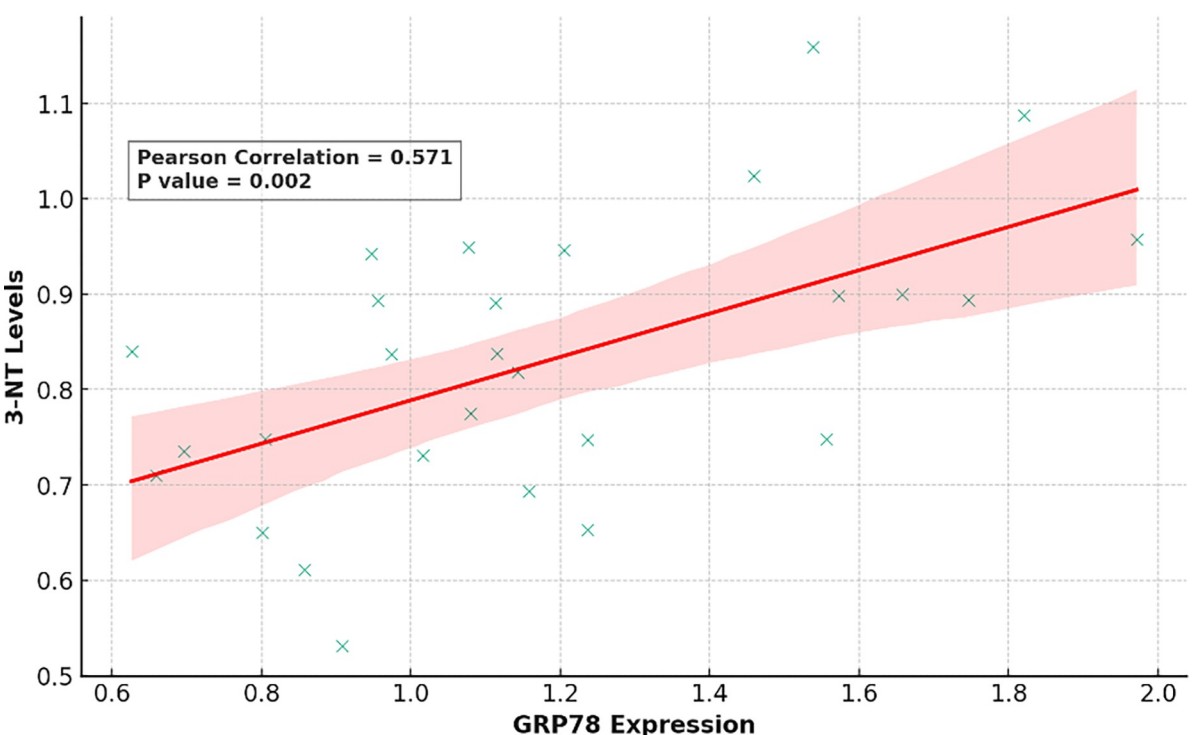

**Fig 6. Pearson linear correlation between 3-NT levels GRP78 relative expression.** Abbreviations: 3-NT, 3-nitrotyrosine; GRP78, 78-kDa glucose-regulated protein.

tissue [33]. In the study by Mainali et al., GRP78 levels were measured one week after inducing MI, which could explain why our results differed. Additionally, exercise appears to reduce GRP78 expression, suggesting that the cardioprotective effect of exercise may be mediated through the reduction of ER stress, which is involved in the intrinsic apoptosis pathway [5, 34].

Furthermore, we discovered that the intensity of nitrosative/oxidative stress, measured by 3-NT, positively correlates with increased expression of GRP78. This indicates that these two components participate in the development of post-MI heart failure (HF).The study by Dickhout et al. observed that 3-NT could colocalize with GRP78 within early atherosclerotic lesions in the walls of arteries, suggesting that nitrosative/oxidative stress and ER stress also contribute to other diseases like coronary artery disease (CAD) [35].

Our findings indicate that iNOS protein levels are increased in cardiac tissue from rats with post-MI HF and with the implementation of HFD, suggesting the occurrence of nitrosative/oxidative stress under both conditions. This observation aligns with other studies demonstrating that the iNOS expression is up-regulated in association with obesity and after MI [19, 35, 36]. Furthermore, iNOS is found to be overexpressed in cardiac tissue in cases of HFpEF [37], which has recently been recognized as a foundation of this disease's pathomechanism [12].

Our study revealed that both nNOS and eNOS levels decrease in post-MI HF in the case of a high-fat and normal-fat diet. Furthermore, eNOS protein levels also decrease after implementing HFD, regardless of MI. Currently, there have been no studies analyzing the impact of HFD on eNOS expression in cardiac tissue. Interestingly, similar to HFD, a high-sugar diet also leads to a reduction in eNOS levels [38]. Moreover, in a model of spontaneously hypertensive rats, eNOS was downregulated explicitly in cardiomyocytes [39]. Similarly, eNOS levels are decreased in hypertrophic cardiomyopathy [40]. However, diabetes does not affect the overall eNOS protein level [41].

We report that both HFD and post-MI HF lead to the elevation of MPO levels. Moreover, studies indicate that even a single high-fat meal can elevate circulating MPO levels and contribute to oxidative stress [42], which could be detrimental if repeated frequently in the context of HF pathogenesis. Elevated MPO levels are present in CAD; also, its high levels may indicate a high risk of acute coronary syndrome (ACS) [43]. Increased level of MPO is related to oxidative stress and inflammatory state in chronic systolic HF. Elevated plasma MPO levels are also associated with an increased likelihood of more advanced HF [18, 44].

We demonstrated that both HFD and MI, as well as the combination of these two factors, significantly increase the activity of the ATF6 axis of the UPR. Interestingly, studies show that ATF6 is critical in protecting the heart after MI. Inhibiting ATF6 activity leads to dilatation of the left ventricle and depression of cardiac function [43]. ATF6 may also regulate the dynamics of CHOP induction [45]. In our study, the levels of CHOP were highest in the control group and both post-MI HF and HFD were associated with the decrease in its levels, which may suggest inhibition of apoptosis related to the ER stress. However, in other studies—HFD or cardiac injury are rather linked with elevated levels of CHOP [46, 47]. Thus, further research is needed to explore this phenomenon.

In our study, we demonstrated that the activity of the IRE1α branch of the UPR was lowest in rats subjected to both HFD and HF simultaneously. This suggests that in the event of an MI, HFD may exacerbate the accumulation of misfolded proteins in cardiomyocytes. Reduced activity of this branch plays a role in the pathogenesis of HFpEF [12]. Until now, no one has demonstrated that a similar reduction in activity can occur in cases where post-MI HF coincides with HFD.

## Limitations

This study primarily relied on Western Blot assays without incorporating PCR assays, a decision driven by the limited budget and the exploratory nature of this project. Additionally, the lack of histopathological evaluation and immunohistochemical verification of the distribution of the studied markers in the left ventricle was significant limitations. The absence of echocardiographic assessment, in particular, was a major issue, as it restricted our ability to evaluate cardiac function. Given the limitations mentioned above, the findings must be interpreted with caution.

## Conclusions

This preliminary study has several limitations, as mentioned above, but provides valuable insights. The findings elucidate the significant impacts of post-MI HF and dietary fat content on cardiac function and stress markers in a rat model. HFD, independently and in combination with post-MI HF, significantly influenced ER stress (as indicated by GRP78 levels), nitrosative/oxidative stress markers (3-NT, iNOS), and other forms of NOS in the heart. Notably, post-MI HF with HFD resulted in the most pronounced alterations in these markers, suggesting exacerbated cardiac dysfunction and stress responses.

The study highlights the interaction between post-MI HF and dietary factors in modulating the UPR pathways (PERK, ATF6, IRE1α) and MPO levels. What is more, HFpEF and the overlap of post-MI HF and HFD may have similarities in pathogenesis. This is a new perspective for future studies.

## Supporting information

**S1 Raw images.**
(PDF)

## Author Contributions

**Conceptualization:** Karol Momot, Jakub Tomaszewski.

**Formal analysis:** Karol Momot, Małgorzata Wojciechowska.

**Funding acquisition:** Maciej Zarębiński.

**Investigation:** Karol Momot, Kamil Krauz, Jakub Dobruch, Tymoteusz Żera.

**Supervision:** Agnieszka Cudnoch-Jędrzejewska.

**Visualization:** Karol Momot, Kamil Krauz.

**Writing – original draft:** Karol Momot, Kamil Krauz.

**Writing – review & editing:** Katarzyna Czarzasta, Małgorzata Wojciechowska.

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
