## [Decision Letter · Decision Letter 0]

14 May 2024

PONE-D-24-15393Post-myocardial infarction heart failure and long-term high-fat diet: cardiac endoplasmic reticulum stress and unfolded protein response in Sprague Dawley rat modelPLOS ONE

Dear Dr. Wojciechowska,

Thank you for submitting your manuscript to PLOS ONE. After careful consideration, we feel that it has merit but does not fully meet PLOS ONE’s publication criteria as it currently stands. Therefore, we invite you to submit a revised version of the manuscript that addresses the points raised during the review process.

Two independent reviewers have evaluated your manuscript. Reviewer #1 states that a substantial and comprehensive revision would be required throughout the paper. Reviewer #2 suggests that the novelty of this experiment is limited and that the authors must provide justification for studying this experimental model.

If you believe you can effectively address all their suggestions, I am open to reconsidering it. Specifically, I would like you to consider the questions raised by Reviewer #1. Additionally, the revised version will need to undergo review by the same reviewers, and the final decision will be based on their assessment.

We look forward to receiving your revised manuscript.

Kind regards,

Marcia B. Aguila, Ph.D.

Academic Editor

PLOS ONE

Journal Requirements:

"This study was financed by a research grant from the Medical University of Warsaw (1MA/2/M/MG/N/23) to [KK]

https://pnitt.wum.edu.pl/en"

Reviewers' comments:

Reviewer's Responses to Questions

**Comments to the Author**

1. Is the manuscript technically sound, and do the data support the conclusions?

Reviewer #1: No

Reviewer #2: Yes

2. Has the statistical analysis been performed appropriately and rigorously? 

Reviewer #1: No

Reviewer #2: Yes

3. Have the authors made all data underlying the findings in their manuscript fully available?

Reviewer #1: No

Reviewer #2: No

4. Is the manuscript presented in an intelligible fashion and written in standard English?

Reviewer #1: No

Reviewer #2: Yes

5. Review Comments to the Author

Reviewer #1: Review PONE-D-24-15393

The study investigated the effect of post-myocardial infarction (MI), heart failure (HF), and high-fat diet HFD on inflammation, nitro-oxidative stress, endoplasmic reticulum (ER) stress, and unfolded protein response (UPR). Twenty-eight male adult Sprague Dawley rats were divided into four groups (NFD vs. HFD) and surgical procedures (sham operation vs. coronary artery ligation to induce MI). The authors concluded that dietary fat content significantly impacts cardiac function and stress markers in a rat model of MI HF. However, several methodological weaknesses and speculative conclusions must be resolved to make the study more scientifically acceptable.

• Adherence to strict dietary standards is a cornerstone of rodent research, ensuring the reproducibility and reliability of results. The dietary pattern of rodents, as outlined in AIN93 (J Nutr 1993;123:1939-51 doi:10.1093/jn/123.11.1939), is a crucial aspect that is unfortunately not detailed in the manuscript. Different types of fat can have varying effects on animal metabolism, and the protein content of the food is also vital for post-surgical recovery. Therefore, I kindly request a table providing a comprehensive breakdown of the ingredients in the control and high-fat diets used in the study.

• What was the difference in energy between the control and HF diets?

• The names of groups should be more explicit and not confusing.

• Page 5 - Fig 1 is mentioned, but we see a table. In line 103, three references are cited to explain the experimental procedures (refs. #24-26). However, these references are a meta-analysis and two reviews that do not provide experimental details.

• Coronary artery ligation in rodents is not always lethal, but many animals die. The manuscript does not inform how many animals were operated on or the survival rate.

• When ligating the coronary artery to cause a myocardial infarction in rodents, the method produces individual lesions; some animals may have an extensive infarction area, but others may not. Evans blue and triphenyl tetrazolium chloride staining can determine infarct areas at sacrifice (a missing item that compromises the study).

• Did the operated animals eat normally? How was the animals' food intake? The information is relevant to understanding the result of body mass loss reported in the article.

• Line 139 –reference #34 used to describe the protein analysis method is a meta-analysis, so…

• I did not find an explanation for what proBNP means (which appears a few times in the text).

• Lines 206-2010 - Nitric oxide (NO) is a gas that degrades quickly. The study did not evaluate NO production and cannot discuss NO levels.

• Comparisons between groups (and markings on graphs) - there are comparisons between groups that are not comparable (e.g., unoperated ND vs. operated HFD) and should not be stated or commented on.

• The statistical analysis should be more explicit and inform where the parametric or non-parametric analysis was performed. Why was the Spearman coefficient not determined for non-parametric correlations?

• I do not understand how the authors talk about 'heart failure' (HF) if the study does not include measuring the animals' blood pressure or a cardiac functional test (ultrasound?). Analysis of left ventricular ejection fraction is essential to understand the effect of post-infarction HFD.

• There are numerous grammatical and spelling errors in the text.

Reviewer #2: This manuscript describes an exploratory study examining the influence of high fat diet on cardiac nitrosative stress and endoplasmic reticulum stress in rat model with post-myocardial infarction heart failure. Overall, the manuscript is well written. Although the novelty of this experiment is limited by the great depth of literature available using other rodent models, this manuscript reports a technically-sound experiment describing the nitrosative and endoplasmic reticulum stress markers in rats. Nonetheless, the manuscript lacks any functional metrics that could support and verify the development of heart failure in this model. The choice of single time-point also limits the information provided by this characterization study whereby it is unclear when does nitrosative stress and endoplasmic reticulum stress occurs with diet and MI-induced heart failure.

Minor comments:

Individual data points should be presented in bar graphs.

6. PLOS authors have the option to publish the peer review history of their article (what does this mean?). If published, this will include your full peer review and any attached files.

Reviewer #1: No

Reviewer #2: No

---

## [Author Response · Author response to Decision Letter 0]

12 Jul 2024

Dear Editor,

 On behalf of all co-authors, I would like to thank the Reviewers for their valuable comments. We have addressed each comment and revised the appropriate sections accordingly. We believe these revisions have significantly improved our manuscript.

Please find our detailed responses to the specific comments below. All modifications in the text are highlighted in yellow.

Sincerely, 

Karol Momot

Małgorzata Wojciechowska

Reviewer #1: 

  The study investigated the effect of post-myocardial infarction (MI), heart failure (HF), and high-fat diet HFD on inflammation, nitro-oxidative stress, endoplasmic reticulum (ER) stress, and unfolded protein response (UPR). Twenty-eight male adult Sprague Dawley rats were divided into four groups (NFD vs. HFD) and surgical procedures (sham operation vs. coronary artery ligation to induce MI). The authors concluded that dietary fat content significantly impacts cardiac function and stress markers in a rat model of MI HF. However, several methodological weaknesses and speculative conclusions must be resolved to make the study more scientifically acceptable.

⁃ Thank you for your thorough review and valuable suggestions. We have addressed your comments and made the necessary revisions, as detailed below.

 • Adherence to strict dietary standards is a cornerstone of rodent research, ensuring the reproducibility and reliability of results. The dietary pattern of rodents, as outlined in AIN93 (J Nutr 1993;123:1939-51 doi:10.1093/jn/123.11.1939), is a crucial aspect that is unfortunately not detailed in the manuscript. Different types of fat can have varying effects on animal metabolism, and the protein content of the food is also vital for post-surgical recovery. Therefore, I kindly request a table providing a comprehensive breakdown of the ingredients in the control and high-fat diets used in the study.

⁃ We have prepared a table with detailed information about the ingredients in the normal-fat and high-fat diets used in the study, based on the manufacturer's specifications. We have also included the prepared table in the text at line 126. Thank you for highlighting this important aspect.

 • What was the difference in energy between the control and HF diets?

⁃ We have also included information about the caloric content of both the control and high-fat diets in the table (line 126). Thank you for your attention to this detail. This type of feed, NFD and HFD, has been used in our laboratory for many years: 

1. Cudnoch-Jedrzejewska A, Gomolka R, Szczepanska-Sadowska E, Czarzasta K, Wrzesien R, Koperski L, Puchalska L, Wsol A. High-fat diet and chronic stress reduce central pressor and tachycardic effects of apelin in Sprague-Dawley rats. Clin Exp Pharmacol Physiol. 2015 Jan;42(1):52-62. doi: 10.1111/1440-1681.12324. PMID: 25311903.

2. Czarzasta K, Cudnoch-Jedrzejewska A, Szczepanska-Sadowska E, Fus L, Puchalska L, Gondek A, Dobruch J, Gomolka R, Wrzesien R, Zera T, Gornicka B, Kuch M. The role of apelin in central cardiovascular regulation in rats with post-infarct heart failure maintained on a normal fat or high fat diet. Clin Exp Pharmacol Physiol. 2016 Oct;43(10):983-94. doi: 10.1111/1440-1681.12617. PMID: 27378063.

3. Czarzasta K, Koperski L, Segiet A, Janiszewski M, Kuch M, Gornicka B, Cudnoch-Jedrzejewska A. The role of high fat diet in the regulation of MAP kinases activity in left ventricular fibrosis. Acta Histochem. 2019 Apr;121(3):303-310. doi: 10.1016/j.acthis.2019.01.010. Epub 2019 Feb 4. PMID: 30733042.

4. Czarzasta K, Wojno O, Zera T, Puchalska L, Dobruch J, Cudnoch-Jedrzejewska A. The influence of post-infarct heart failure and high fat diet on the expression of apelin APJ and vasopressin V1a and V1b receptors. Neuropeptides. 2019 Dec;78:101975. doi: 10.1016/j.npep.2019.101975. Epub 2019 Oct 15. PMID: 31645268.

5. Wojno O, Czarzasta K, Puchalska L, Kowalczyk M, Cudnoch-Jedrzejewska A. Central interaction between the apelinergic and vasopressinergic systems in the regulation of the haemodynamic parameters in rats maintained on a high-fat diet. Clin Exp Pharmacol Physiol. 2020 Dec;47(12):1902-1911. doi: 10.1111/1440-1681.13381. Epub 2020 Aug 16. PMID: 32687615.

 Normal-fat Diet High-fat Diet

Energy (kcal per 100g) 286 382

Carbohydrates (g per 100g) 60.0 35.5

Proteins (g per 100g) 17.4 17.1

Fats (g per 100g) 3.2 28.0

Saturated (%) 13 49

Unsaturated (%) 37 44

Polyunsaturated (%) 50 7

Raw ash (g per 100g) 12 12

Water (g per 100g) 12 12

 • The names of groups should be more explicit and not confusing.

⁃ We have changed the group names from HFD HF to HFD-MI and NFD HF to NFD-MI to make them more explicit and less confusing. All the changed names have been highlighted in yellow. Thank you for your suggestion.

 • Page 5 - Fig 1 is mentioned, but we see a table. 

⁃ At this point in the text, both Figure 1 and Table 1 should be included. Table 1 is included in the text, and following the author guidelines, figures should not be placed within the main text. Figure 1 is included in the attachments.

• In line 103, three references are cited to explain the experimental procedures (refs. #24-26). However, these references are a meta-analysis and two reviews that do not provide experimental details.

⁃ The appropriate references are now #29-31, which were initially listed at the bottom of the reference list (Line 104). Apologies for the error.

 • Coronary artery ligation in rodents is not always lethal, but many animals die. The manuscript does not inform how many animals were operated on or the survival rate.

⁃ We have added information about the number of animals operated on and the survival rates in the main text:

⁃ Sham-operated and High-fat Diet: 10 rats operated, 6 survived

⁃ Coronary artery ligation and High-fat Diet: 20 rats operated, 10 survived

⁃ Sham-operated and Normal-fat diet: 10 rats operated, 7 survived

⁃ Coronary artery ligation and Normal-fat diet: 15 rats operated, 9 survived

Please see Table 1, line 110.

 • When ligating the coronary artery to cause a myocardial infarction in rodents, the method produces individual lesions; some animals may have an extensive infarction area, but others may not. Evans blue and triphenyl tetrazolium chloride staining can determine infarct areas at sacrifice (a missing item that compromises the study).

⁃ We measured the infarct area using the planimetric method: the HFD-MI group had a significantly larger infarction area than the NFD-MI group (44.57 ± 3.44% vs. 25.29 ± 0.75%, P< 0.001). We did not use Evans blue and triphenyl tetrazolium chloride staining because post-euthanasia, we collected the heart (both right and left ventricles) for gene expression analysis using RT-PCR or to estimate protein levels using Western Blot or ELISA. Staining the hearts could have affected the results of these biochemical analyses or rendered them impossible. Therefore, we opted for the planimetric method to assess the infarct scar.

 • Did the operated animals eat normally? How was the animals' food intake? The information is relevant to understanding the result of body mass loss reported in the article.

⁃ We did not measure the food intake of the operated animals. However, we did assess fat tissue masses, which can be found in the articles cited in our main text:

1. Czarzasta K, Cudnoch-Jedrzejewska A, Szczepanska-Sadowska E, et al. The role of apelin in central cardiovascular regulation in rats with post-infarct heart failure maintained on a normal fat or high fat diet. Clin Exp Pharmacol Physiol. 2016;43(10):983-94. https://doi.org/10.1111/1440-1681.12617

2. Czarzasta K, Wojno O, Zera T, et al. The influence of post-infarct heart failure and high fat diet on the expression of apelin APJ and vasopressin V1a and V1b receptors. Neuropeptides. 2019;78:101975. https://doi.org/10.1016/j.npep.2019.101975

 • Line 139 –reference #24 used to describe the protein analysis method is a meta-analysis, so…

⁃ The situation is the same as in line 103. We apologize for the mistake. The correct citation should be #29. Thank you for your attention to detail. Please see line 143.

 • I did not find an explanation for what proBNP means (which appears a few times in the text).

⁃ We have added an explanation for the abbreviation proBNP in the text. Please see line 132-133. Thank you for pointing this out.

 • Lines 206-2010 - Nitric oxide (NO) is a gas that degrades quickly. The study did not evaluate NO production and cannot discuss NO levels.

⁃ We agree with the Reviewer’s opinion. However, in our study, we analyzed the levels of synthases, which are relatively stable proteins and do not degrade as quickly as NO. These can be analyzed using Western blot.

 • Comparisons between groups (and markings on graphs) - there are comparisons between groups that are not comparable (e.g., unoperated ND vs. operated HFD) and should not be stated or commented on.

⁃ Thank you for your observation. We have corrected the figures and the text to ensure only comparable groups, which differ by a single variable (either the type of procedure or the diet, but not both simultaneously), are stated and commented on..

 • The statistical analysis should be more explicit and inform where the parametric or non-parametric analysis was performed. Why was the Spearman coefficient not determined for non-parametric correlations?

⁃ We did not use the Spearman coefficient because each analyzed data point followed a normal distribution and exhibited a linear relationship. Therefore, we used only the Pearson method. Additionally, upon closer analysis, we confirmed that all quantitative data were analyzed using one-way ANOVA, and none with the Kruskal-Wallis test. We have added a note about this in the methodology section. See lines 158-164. Thank you for your suggestion and apologies for the confusion.

 • I do not understand how the authors talk about 'heart failure' (HF) if the study does not include measuring the animals' blood pressure or a cardiac functional test (ultrasound?). Analysis of left ventricular ejection fraction is essential to understand the effect of post-infarction HFD.

⁃ We assessed the infarct area using the planimetric method and measured NT-proBNP levels in plasma to determine if the rats had developed post-infarction heart failure. Thank you for highlighting the importance of these measurements.

⁃ In a separate study, we evaluated hemodynamic parameters, including left ventricular end-diastolic pressure (LVEDP), using invasive catheterization in both infarcted and non-infarcted rats on high-fat or standard diets: 

⁃ Czarzasta K, Cudnoch-Jedrzejewska A, Szczepanska-Sadowska E, et al. The role of apelin in central cardiovascular regulation in rats with post-infarct heart failure maintained on a normal fat or high fat diet. Clin Exp Pharmacol Physiol. 2016;43(10):983-94. https://doi.org/10.1111/1440-1681.12617). 

 • There are numerous grammatical and spelling errors in the text.

⁃ Thank you for pointing this out. We have thoroughly reviewed the text again and corrected all grammatical and spelling errors.

Reviewer #2: 

 This manuscript describes an exploratory study examining the influence of high fat diet on cardiac nitrosative stress and endoplasmic reticulum stress in rat model with post-myocardial infarction heart failure. Overall, the manuscript is well written. 

⁃ Thank you very much for taking the time to review our manuscript. We have addressed your comments below.

Although the novelty of this experiment is limited by the great depth of literature available using other rodent models, this manuscript reports a technically-sound experiment describing the nitrosative and endoplasmic reticulum stress markers in rats. 

⁃ Thank you for your feedback. As this was a preliminary study and the results were very interesting, we decided to submit this article to the journal. To date, no one has analyzed nitrosative stress in the context of myocardial infarction and a high-fat diet. It is known that this stress is exacerbated in heart failure with preserved ejection fraction. Publishing these results will lay the foundation for further research and inspire more researchers to explore this topic.

Nonetheless, the manuscript lacks any functional metrics that could support and verify the development of heart failure in this model. 

⁃ We assessed the infarct area using planimetric methods and measured NT-proBNP levels in plasma to determine if the rats had developed post-infarction heart failure. In a separate study, we also evaluated hemodynamic parameters, including LVEDP (left ventricular end-diastolic pressure) using invasive catheterization in both infarcted and non-infarcted rats on high-fat or standard diets (refer to Table in Experiment 1 from DOI: 10.1111/1440-1681.12617).

The choice of single time-point also limits the information provided by this characterization study whereby it is unclear when does nitrosative stress and endoplasmic reticulum stress occurs with diet and MI-induced heart failure.

⁃ Thank you for your insightful feedback. We would like to emphasize that these are preliminary studies, and extending the study to include additional time points is challenging due to the high mortality rate of rats following the ligation of the left coronary artery. Specifically, we observed the following survival rates in our study (See Table 1, line 110):

⁃ Sham-operated and High-fat Diet: 10 rats operated, 6 survived

⁃ Coronary artery ligation and High-fat Diet: 20 rats operated, 10 survived

⁃ Sham-operated and Normal-fat diet: 10 rats operated, 7 survived

⁃ Coronary artery ligation and Normal-fat diet: 15 rats operated, 9 survived 

We chose to include one control group and three experimental groups in our study. This design allowed us to use two-way ANOVA analysis to evaluate the impact of different factors on various parameters of stress and inflammation

---

## [Decision Letter · Decision Letter 1]

18 Jul 2024

PONE-D-24-15393R1Post-myocardial infarction heart failure and long-term high-fat diet: cardiac endoplasmic reticulum stress and unfolded protein response in Sprague Dawley rat modelPLOS ONE

Dear Dr. Wojciechowska,

Thank you for submitting your manuscript to PLOS ONE. After careful consideration, we feel that it has merit but does not fully meet PLOS ONE’s publication criteria as it currently stands. Therefore, we invite you to submit a revised version of the manuscript that addresses the points raised during the review process.

The reviewer #2 approved the authors' responses but requested a comment in the conclusion indicating that this study is preliminary. However, the reviewer #1 did not accept the authors' responses and insists on them being addressed adequately. As the academic editor, I will grant the authors another opportunity, and the responses will be forwarded to reviewer #1 for evaluation.

We look forward to receiving your revised manuscript.

Kind regards,

Marcia B. Aguila, Ph.D.

Academic Editor

PLOS ONE

Reviewers' comments:

Reviewer's Responses to Questions

**Comments to the Author**

1. If the authors have adequately addressed your comments raised in a previous round of review and you feel that this manuscript is now acceptable for publication, you may indicate that here to bypass the “Comments to the Author” section, enter your conflict of interest statement in the “Confidential to Editor” section, and submit your "Accept" recommendation.

Reviewer #1: (No Response)

Reviewer #2: All comments have been addressed

2. Is the manuscript technically sound, and do the data support the conclusions?

Reviewer #1: Partly

Reviewer #2: Yes

3. Has the statistical analysis been performed appropriately and rigorously? 

Reviewer #1: No

Reviewer #2: Yes

4. Have the authors made all data underlying the findings in their manuscript fully available?

Reviewer #1: Yes

Reviewer #2: Yes

5. Is the manuscript presented in an intelligible fashion and written in standard English?

Reviewer #1: Yes

Reviewer #2: Yes

6. Review Comments to the Author

Reviewer #1: Review PONE-D-24-15393R1

I thank the authors for reviewing the manuscript and considering previous comments. However, there are still some issues that need to be resolved:

Lines 137-138 "The infarction surface was measured planimetrically". Planimetry is a method for estimating areas (just like a scale measures mass). The issue with evaluating the infarcted area in the left ventricle is that the manuscript does not have any information on how this was done. As mentioned in the first review, Evans blue and triphenyl tetrazolium chloride staining is generally used to determine infarct areas at sacrifice. However, the authors responded that they did not do it this way but by planimetry (?). OK, but how did the authors select the region of infarction? Were histological sections made from the left ventricle and a dye used? How many cuts, and at what levels of the left ventricle? How did the authors perform the fractionation of the ventricle to estimate the infarction area? Does planimetry use point counting, image analysis, or other techniques? It is essential to highlight that infarction is a three-dimensional issue; planimetry alone (which determines the areas of infarction in sections) does not provide information on the volume of the injured myocardium. Unfortunately, this result should be removed from the article because it does not make sense scientifically.

Page 8 (statistical analysis) – It is problematic that such a small sample was normally distributed for all parameters analyzed (perhaps the authors should review this). It is not understood to do a one-way ANOVA and then a two-way ANOVA. What software was used in the study? Bio-render and AI-based tools are not acceptable software for statistics.

Finally, the lack of a functional study of the post-infarct heart and estimation of left ventricular ejection fraction significantly reduces the contribution that could be made by the experiment and is a significant limitation of the study.

Reviewer #2: Authors have addressed all my previous comments. They have acknowledge that this is a preliminary study and it would be ideal that they mention this in the discussion and conclusion.

7. PLOS authors have the option to publish the peer review history of their article (what does this mean?). If published, this will include your full peer review and any attached files.

Reviewer #1: No

Reviewer #2: No

---

## [Author Response · Author response to Decision Letter 1]

24 Jul 2024

TO REVIEWER 1:

Dear Reviewer,

 Thank you for your feedback and suggestions. We have carefully considered your comments. Below, we provide our responses point by point:

1. As requested, we have decided to remove the section on planimetry from the manuscript.

 However, to provide a clear understanding of our methodology, we would like to include the following detailed explanation only in this correspondence: 

 After euthanasia, the heart was excised from the thorax, and the left ventricle, including the septum, was separated from the right ventricle. Both ventricles were weighed. Infarct size was determined planimetrically as previously described [Leenen et al. 1999; Brain “ouabain” and angiotensin II contribute to cardiac dysfunction after myocardial infarction; doi.org/10.1152/ajpheart.1999.277.5.H1786], with some modifications [Dobruch et al. 2009; Enhanced involvement of brain vasopressin V1 receptors in cardiovascular responses to stress in rats with myocardial infarction, doi.org/10.1080/10253890500456287]. The left ventricle was cut along the longitudinal axis and placed flat on a transparent plastic sheet. The infarcted areas on both the inner (endocardial) and outer (epicardial) surfaces of the ventricle were outlined. The average of these two measurements was then calculated and expressed as a percentage of the total left ventricle wall size.

2. We acknowledge your point that one-way ANOVA is inappropriate. In response, we have removed all references to it from the methodology and results sections of the manuscript. 

 We have retained the two-way ANOVA analysis to examine the effects of diet (Normal-fat Diet vs. High-fat Diet) and procedure (Sham operation vs. Myocardial infarction) and their interaction. The revised methodology section now reads: 

 "A two-way ANOVA was conducted to examine the effects of diet (Normal-fat Diet vs. High-fat Diet) and procedure (Sham operation vs. Myocardial infarction), as well as their interaction. Post-hoc comparisons were performed using Tukey’s HSD test to identify significant differences between groups."

3. Regarding your concerns about the normality of data distribution, we conducted Shapiro-Wilk tests during statistical analysis; the results are provided in the table below. It is worth noting that each Western blot measurement was repeated three times, and the final result was calculated as the average of these repetitions. 

 Despite a single instance where the Shapiro-Wilk test indicated non-normality (p < 0.05) for "HFD MI," the use of a two-way ANOVA is justified. Levene's test for homogeneity of variances yielded a p-value of 0.09195, indicating no significant difference in variances between groups. ANOVA is robust to minor deviations from normality, especially with similar sample sizes. Therefore, acknowledging the limitation, two-way ANOVA remains appropriate for this analysis.

4. Regarding your question about the software used in the study, we have clarified this point in the manuscript. Statistical analysis was carried out using Statistica software, version 13.3. This information has been added to the "Statistical Analysis" section of the main text.

5. We acknowledge your point regarding the lack of a functional study of the post-infarct heart and the estimation of left ventricular ejection fraction. 

 Unfortunately, the study has already been conducted, and we cannot address this limitation. 

 However, we believe that our study, as preliminary research, provides valuable insights and perspectives for future investigations. We have emphasized this limitation once more in the "Limitations" section at the end of the article to ensure that readers interpret the findings with caution.

6. We have made the necessary revisions to the text in response to your comments and have emphasized that this study is preliminary research both in the discussion and conclusion sections. 

For your review, the changes have been made using the track changes mode.

 We sincerely thank both reviewers for the valuable and scientific discussion and appreciate all the effort put into the thorough review. We hope this clarification addresses your concerns.

Sincerely,

Karol Momot and colleagues

Diet Procedure Shapiro Wilk P-value (GRP/actin)

HFD MI 0.823293

HFD SO 0.590984

NFD MI 0.565416

NFD SO 0.368853

Diet Procedure Shapiro Wilk P-value (mean ATF6c/ATF6)

HFD MI 0.046620

HFD SO 0.612515

NFD MI 0.720524

NFD SO 0.590896

Diet Procedure Shapiro Wilk P-value (PERK/actin)

HFD MI 0.094379

HFD SO 0.566535

NFD MI 0.784923

NFD SO 0.820542

Diet Procedure Shapiro Wilk P-value (CHOP/actin)

HFD MI 0.419551

HFD SO 0.408671

NFD MI 0.258975

NFD SO 0.460599

Diet Procedure Shapiro Wilk P-value (nNOS/actin)

HFD MI 0.782037

HFD SO 0.589341

NFD MI 0.758002

NFD SO 0.746428

Diet Procedure Shapiro Wilk P-value (eNOS/actin)

HFD MI 0.464793

HFD SO 0.696505

NFD MI 0.573740

NFD SO 0.181428

Diet Procedure Shapiro Wilk P-value (NT/actin)

HFD MI 0.288189

HFD SO 0.166167

NFD MI 0.364409

NFD SO 0.909621

Diet Procedure Shapiro Wilk P-value (MPO/actin)

HFD MI 0.200157

HFD SO 0.861766

NFD MI 0.108960

NFD SO 0.918455

Diet Procedure Shapiro Wilk P-value (iNOS/actin)

HFD MI 0.118528

HFD SO 0.884183

NFD MI 0.923192

NFD SO 0.117495

Diet Procedure Shapiro Wilk P-value (pIRE/IRE)

HFD MI 0.253621

HFD SO 0.416927

NFD MI 0.060167

NFD SO 0.238902

Diet Procedure Shapiro Wilk P-value (NT-proBNP)

HFD MI 0.656257

HFD SO 0.133989

NFD MI 0.427857

NFD SO 0.194803

Diet Procedure Shapiro Wilk P-value (body mass)

HFD MI 0.624347

HFD SO 0.255299

NFD MI 0.371023

NFD SO 0.560306

TO REVIEWER 2:

Dear Reviewer,

 Thank you for your feedback and suggestions. We have made the necessary revisions to the text in response to reviews and have emphasized that this study is preliminary research in both the discussion and conclusion sections. 

For your review, the changes have been made using the track changes mode.

 We sincerely thank both reviewers for the valuable and scientific discussion. Your input has been greatly appreciated and has significantly contributed to the development of our study. We hope this clarification addresses your concerns.

Sincerely,

Karol Momot and colleagues

---

## [Decision Letter · Decision Letter 2]

30 Jul 2024

Post-myocardial infarction heart failure and long-term high-fat diet: cardiac endoplasmic reticulum stress and unfolded protein response in Sprague Dawley rat model

PONE-D-24-15393R2

Dear Dr. Wojciechowska,

We’re pleased to inform you that your manuscript has been judged scientifically suitable for publication and will be formally accepted for publication once it meets all outstanding technical requirements.

Kind regards,

Marcia B. Aguila, Ph.D.

Academic Editor

PLOS ONE

Additional Editor Comments (optional):

Reviewers' comments:

Reviewer's Responses to Questions

**Comments to the Author**

1. If the authors have adequately addressed your comments raised in a previous round of review and you feel that this manuscript is now acceptable for publication, you may indicate that here to bypass the “Comments to the Author” section, enter your conflict of interest statement in the “Confidential to Editor” section, and submit your "Accept" recommendation.

Reviewer #1: All comments have been addressed

2. Is the manuscript technically sound, and do the data support the conclusions?

Reviewer #1: Yes

3. Has the statistical analysis been performed appropriately and rigorously? 

Reviewer #1: Yes

4. Have the authors made all data underlying the findings in their manuscript fully available?

Reviewer #1: Yes

5. Is the manuscript presented in an intelligible fashion and written in standard English?

Reviewer #1: Yes

6. Review Comments to the Author

Reviewer #1: The authors accepted my suggestions and placed a limitation on the study. The study is preliminary and should be continued in the future. No further comments.

7. PLOS authors have the option to publish the peer review history of their article (what does this mean?). If published, this will include your full peer review and any attached files.

Reviewer #1: No

---

## [Editor Report · Acceptance letter]

1 Aug 2024

PONE-D-24-15393R2 

PLOS ONE

Dear Dr. Wojciechowska, 

I'm pleased to inform you that your manuscript has been deemed suitable for publication in PLOS ONE. Congratulations! Your manuscript is now being handed over to our production team.

Kind regards, 

on behalf of

Dr. Marcia B. Aguila 

Academic Editor

PLOS ONE